# May Women with a Negative Co-Test at First Follow-Up Visit Return to 3-Year Screening after Treatment for Cervical Intraepithelial Neoplasia?

**DOI:** 10.3390/ijerph20064739

**Published:** 2023-03-08

**Authors:** Finn Egil Skjeldestad, Sveinung Wergeland Sørbye

**Affiliations:** 1Research Group Epidemiology of Chronic Diseases, Department of Community Medicine, UiT The Arctic University of Norway, 9037 Tromsø, Norway; 2Department of Clinical Pathology, University Hospital of North Norway, 9038 Tromsø, Norway

**Keywords:** cervical intraepithelial neoplasia, residual disease, recurrent disease, adherence, guidelines, treatment

## Abstract

Background: The Norwegian Cervical Cancer Screening Programme recommends that women treated for cervical intraepithelial neoplasia (CIN) only be returned to 3-year screening after receiving two consecutive negative co-tests, 6 months apart. Here we evaluate adherence to these guidelines and assessed the residual disease, using CIN3+ as the outcome. Methods: This cross-sectional study comprised 1397 women, treated for CIN between 2014 and 2017, who had their cytology, HPV, and histology samples analyzed by a single university department of pathology. Women who had their first and second follow-up at 4–8 and 9–18 months after treatment were considered adherent to the guidelines. The follow-up ended on 31 December 2021. We used survival analysis to assess the residual and recurrent CIN3 or worse among women with one and two negative co-tests, respectively. Results: 71.8% (1003/1397) of women attended the first follow-up 4–8 months after treatment, and 38.3% were considered adherent at the second follow-up. Nearly 30% of the women had incomplete follow-up at the study end. None of the 808 women who returned to 3-year screening after two negative co-tests were diagnosed with CIN3+, whereas two such cases were diagnosed among the 887 women who had normal cytology/ASCUS/LSIL and a negative HPV test at first follow-up (5-year risk of CIN3+: 0.24, 95%, CI: 0.00–0.57 per 100 woman-years). Conclusions: The high proportion of women with incomplete follow-up at the end of the study period requires action. The risk of CIN3+ among women with normal cytology/ASCUS/LSIL and a negative HPV test at first follow-up is indicative of a return to 3-year screening.

## 1. Introduction

Many countries started using co-tests, i.e., cytology in combination with human papillomavirus (HPV) testing, in the follow-up of women treated for cervical intraepithelial neoplasia (CIN), even before the implementation of primary HPV testing in cervical cancer screening. The main differences across countries are the timing of the first post-treatment follow-up and the number negative co-tests required before women can return to 3-year screening in the general screening program.

The Norwegian Cervical Cancer Screening Programme (NCCSP) recommends that women treated for CIN only be returned to 3-year screening after receiving two consecutive negative co-tests, 6 months apart: the first at 6 and the second (repeat co-test) at 12 months after treatment [1]. Sweden recommends returning to a 3-year screening interval after one negative co-test at the 6-month follow-up [2]. Recommendations in the United States and Australia specify two negative co-tests at 12 and 24 months after treatment, or two [3] or three [4] consecutive negative co-tests 12 months apart later. The United Kingdom recommends a first follow-up at 6 months after treatment and a second at 12 or 18 months, depending upon resection margins, if the first co-test is negative [5]. While Denmark recommends free resection margins and a negative co-test at 6 months post-treatment before returning women to 3-year screening [6], Finland only uses these criteria for women treated for CIN1 who had a negative co-test 6 months after treatment [7].

The most worrisome post-treatment scenario is a diagnosis of occult cancer that was missed at primary treatment. In the many published studies, few cases of cancer have been reported during the first year of follow-up [8]. The other challenging issue is the persistence of HPV infections [9]. The closer to the treatment the first follow-up is scheduled, the more likely it is that the HPV portion of the co-test will be positive. As HPV infections wane over time [9], a 12-month interval before the first follow-up will reduce over-diagnosis and unnecessary follow-up due to false-positive HPV tests in the presence of normal cytology or minor cytological abnormalities. As most studies on this topic are short-term, risk assessments of new guidelines in longer duration prospective studies must be carried out before a more global follow-up regimen can be agreed upon. 

In the present study, we evaluated adherence to the national follow-up guidelines after treatment for CIN and assessed the residual disease using CIN3+ as the outcome in an historical prospective case series design. 

## 2. Material and Methods 

In 2015, selected counties in Norway began to use either cytology or HPV testing as a primary cervical cancer screening method among women aged from 34 to 69 years; women were randomized by birth date to determine the screening method. The remaining counties used cytology as the primary screening method in this age group [10], and cytology remained the primary screening method for all women aged from 25 to 33 years in the country. Women with high-grade squamous intraepithelial lesions (HSIL), atypical squamous cells HSIL cannot be excluded, glandular cells of undetermined significance, adenocarcinoma in situ (ACIS), and cervical cancer are referred directly to colposcopy/biopsy. The follow-up of women with unsatisfactory cytology, atypical squamous cells of undetermined significance (ASCUS) and low-grade squamous intraepithelial lesion (LSIL), is determined by reflex HPV testing. In 2014, the NCCSP recommended that women treated for CIN be followed up with co-tests at 6 and 12 months after treatment, free of charge, reimbursed by the public Norwegian social security system, at the referral practitioner’s office either in general practice or in private gynecologic settings [1].

The Department of Pathology, University Hospital of North Norway (UHNN), Tromsø, performs both cytological and histological assessments for all residents of Troms and Finnmark counties. As in all laboratories in Norway, our department assesses cytology samples according to the Bethesda system [11], and histology samples according to the World Health Organization criteria for CIN1-3, ACIS, and cervical cancer [12]. Our department utilized cytology as the primary screening method until 2019, when we began the use of either cytology or HPV testing based on the randomized birth date for women aged 34–69 years [10].

The UHNN’s clinical database, SymPathy, contains information on screening history (cytology, HPV, and histology results), treatment, and follow-up. Using that database, we identified 1424 residents of Troms and Finnmark counties who received treatment for abnormalities in the cervix uteri from 1 January 2014 through to 31 December 2017. After exclusion of women with a diagnosis of cervical cancer in biopsies/cone specimens (n = 27), 1397 women were included in our analyses.

We categorized age into six groups, in line with the age differentiation in the NCCSP (20–24, 25–29, 30–33, 34–49, 50–59, 60–69, and 70–84 years). Histological diagnoses in biopsies and cone specimens were recorded as normal, CIN1, CIN2, CIN3 (including ACIS), and cervical cancer. Resection margins were categorized as free or not free, with the latter category including missing and inconclusive assessments.

We applied a pragmatic approach when analyzing the follow-up, expanding the window for the first and second follow-up to 4–8 months and 9–18 months after treatment, respectively; women who attended both follow-ups within these windows were considered adherent. Women who attended no follow-up, one follow-up, the first follow-up before or within the expanded window but the second follow-up after 18 months, or the first and second follow-up after 18 months, were categorized as non-adherent. Women scheduled for biopsy at the second follow-up (based on the results from the first follow-up) were considered adherent if their biopsy was collected within 6 months of the first follow-up. Women scheduled for biopsy at the second follow-up who had their biopsy taken after 6 months or received a repeat co-test instead were considered non-adherent. If a woman had a cytology sample and a biopsy collected at the same follow-up visit, the histological outcome was used. The follow-up ended on 31 December 2021, when analyses were halted to allow 48 to 96 months of observation post-treatment for all participants without a definitive outcome.

We defined residual disease as histologically confirmed CIN2+. Recurrent disease was defined as a histologically confirmed diagnosis of CIN2+ after two consecutive, negative co-tests. We classified women awaiting treatment, colposcopy/biopsy, or further follow-up for abnormal cytology/HPV results at study end as having "incomplete follow-up". The post-treatment follow-up time was calculated as the time in months between the treatment and the highest histological outcome of CIN2+, the date of the last follow-up visit, the date at which the woman was returned to 3-year screening, or the study end (31 December 2021), depending upon the outcome.

HPV DNA was assessed from liquid-based samples (ThinPrep) by the Cobas 4800 (Roche Molecular Diagnostics) system, in accordance with the national recommendations [13]. We categorized HPV types hierarchically as HPV16 including co-infections, HPV18/not HPV16, and other HPV types/not HPV16/18. We utilized the most recent HPV test results available 24 months before treatment (pre-treatment HPV results).

All analyses were performed in SPSS version 27.0 with a Chi-square test, Fisher’s exact test, and survival analyses. The *p*-values < 0.05 were considered statistically significant.

The Regional Committee for Medical and Health Research Ethics, North Norway, has evaluated the protocol as a quality assurance study fulfilling the requirements for data protection procedures within the department (2015/2479/REK Nord). The Patient Ombudsman, UHNN, Tromsø, approved the study start.

## 3. Results

The mean age at treatment for CIN was 36.2 years (range 19–85 years). Most women (92%) were treated for CIN2 (52%) or 3 (40%) (Table 1). Thirteen cases with normal histology (0.9%) and 98 cases (7%) with CIN1 were also treated due to repetitive abnormal cytology with intermittent positive HPV tests over time. In the subsequent analysis, the 13 normal histology cases were combined with CIN1 cases. The proportion of women with CIN1 increased with increasing age, while the proportion with CIN2 decreased from 62% in the youngest age groups to 45% in the age groups 50 to 69 and 70 to 84 years. For CIN3, the proportion of cases was highest in the age groups 25–29 and 30–33 years (44%). Approximately 50% of women treated for CIN showed disease progression within 1 year of their first abnormal cytology result (Table 1). The mean time from the first abnormal cytology result to CIN2+ decreased from 62 months in 2014 to 42 months in 2017. Five percent of the women had one or more previous CIN treatments, evenly distributed among women with residual and recurrent disease following their previous treatment (Table 1). The proportion of women with pre-treatment HPV results increased, reaching 88% in 2017. A larger proportion of women with high-grade cytological lesions were not tested for HPV, as these cases receive treatment without it. Despite this observation, HPV16 was most prevalent in CIN3 cases, while HPV types other than 16/18 were most prevalent in CIN1 and CIN2 cases. Overall, 2.9% of the women were HPV-negative, more CIN1 than CIN2 and CIN3 cases (Table 1).

In the study sample, 71.8% (1003/1397) of the women attended the first follow-up within 4–8 months, whereas 30 (2.1 %), 276 (19.8%), 69 (4.9%), and 19 (1.4%) did not attend, attended at 1–3 months, 9–18 months, and 19–76 months post-treatment, respectively. Of the 1003 women who attended the first follow-up within 4–8 months, 795 were referred for a repeat co-test and 208 to colposcopy/biopsy, among whom 169 had their biopsy collected within 6 months of the first follow-up. Among the women referred for a repeat co-test, 368 attended their second follow-up within 18 months of the treatment. In total, 38.3% women were considered adherent; 12.1% (169/1397) underwent colposcopy/biopsy, and 26.3% (368/1397) received a repeat co-test. Another 357 women referred for a repeat co-test attended the second follow-up 19 or more months post-treatment, more than 10 months after their first follow-up. Regardless of the attendance at the first follow-up, the women referred to colposcopy/biopsy at the second follow-up (79.6% (95% confidence interval [CI]: 74.9–84.2) (230/289)) had significantly better adherence to the second follow-up than the women referred for a repeat co-test (49.5% (95%, CI: 46.5–52.5) (524/1059)). Similar results were found for the women who did not attend the second follow-up for colposcopy/biopsy and the repeat co-test (1.0% (95%, CI: 0.0–2.1) vs. 6.0 (95%, CI: 4.5–7.4)). Overall, 2.1% (30/1397) and 4.7% (66/1397) of the cases were lost to follow-up at the first and second follow-up, respectively.

At the end of the follow-up period (48–96 months after treatment), 5.4% (n = 76) and 3.2% (n = 45) of the women were diagnosed with residual CIN2 and CIN3, respectively, whereas 0.5% (n = 7) were diagnosed with invasive cervical cancer, and 33.1% (n = 461) had either no or incomplete follow-up (Table 2).

In total, 808 (57.8%) women returned to 3-year screening. At 33 months of follow-up, 50.3% (95%, CI: 47.6–53.0) of the study population had been returned to 3-year screening. This proportion increased from 45.1% at 24 months to 57.5% at 60 months of follow-up (Figure 1). Similarly, the cumulative proportions of residual CIN2+ increased from 5.4% (95%, CI: 4.3–6.5) to 7.1% (95%, CI: 5.9–8.3) at 24 and 60 months of follow-up, respectively (Figure 1). The corresponding figures for CIN2 were 3.1% (95%, CI: 2.3–3.9) to 4.2% (95%, CI: 3.2–5.2), and 2.3% (95%, CI: 1.6–3.0) to 2.8% (95%, CI: 2.0–3.6) for CIN3+ cases, respectively. Most residual CIN2 and CIN3+ cases were diagnosed before 48 months of follow-up (89% and 87%, respectively). For the women with incomplete follow-up, the cumulative proportion of cases increased from 18.1% with the last follow-up within 24 months post-treatment, to 29.5% (95%, CI: 27.0–32.0) within 60 months (Figure 1).

There were seven cases of residual cervical cancer. The age at treatment for these cases ranged from 27 to 67 years. The highest histology in cone specimen or biopsies was CIN3 in six cases, and resection margins were assessed as “free” in three of these seven cases. At the first follow-up, most cases showed HSIL cytology, and all were HPV-positive, most for HPV16. All these women adhered to the follow-up guidelines. Two women with reconization received a cancer diagnosis at follow-up visits after their most recent treatment (Table 3).

Regarding recurrence, the highest histology among the 808 cases returned to screening was CIN2 (two cases), which was diagnosed 24 and 60 months after returning to screening, respectively, 48 and 84 months post-treatment. Among the 887 women with normal cytology/ASCUS/LSIL and a negative HPV outcome at the first follow-up, there were 10 CIN2 and two CIN3 cases identified before the end of the study period. The two CIN3 cases were diagnosed 24 and 51 months post-treatment, while the 10 CIN2 cases were diagnosed 12 to 84 months post-treatment.

Among the 1201 women with a valid co-test at the first follow-up, those with a negative HPV test and normal cytology/ASCUS/LSIL had a 0.11% (95%, CI: 0.00–0.33) and a 0.24% (95%, CI: 0.00–0.57) cumulative 3- and 5-year risk of CIN3+, respectively. The women with a positive HPV test and normal cytology had a 2.4-fold increased risk of CIN3+ at the 5-year follow-up. All other cytology/HPV combinations had cumulative 5-year risks above 4% (Table 4).

In the logistic regression analysis of the resection margins, the months between the first abnormal cytology result and treatment, and the stratified variable of cytology and the HPV test outcomes predicted CIN3+, while age, highest histology outcome, and previous treatment for CIN did not predict CIN3+ (data not shown). We found a highly significant association between HPV positivity and not free resection margins. However, none of these findings added any information that improved the model beyond the co-test outcomes, as only two of the 43 CIN3+ cases had free resection margins and a negative co-test at the first follow-up.

## 4. Discussion

Even with an expanded post-treatment follow-up window, adherence to the recommended two follow-up visits within 18 months of the CIN treatment was low (38.3%). Challenges included the large proportion of women (nearly 30% at 5-year follow-up) with an unresolved clinical situation, continuous visits with inconsistent HPV positivity, and intermittent normal/abnormal cytological lesions/normal or CIN1 histology. Residual CIN2+ increased continuously, reaching 7.1% at the 5-year follow-up. All seven cervical cancer cases were diagnosed following a positive co-test at the first follow-up. Fifty percent of the study population returned to 3-year screening after two consecutive negative co-tests at 33 months of follow-up, increasing to 60% more than 60 months post-treatment. Our data suggest that it is safe to return women with normal/ASCUS/LSIL cytology and a negative HPV test at first the follow-up to 3-year screening.

Adherence to the post-treatment follow-up guidelines was lower in the present study than from the same hospitals during 2006–2011 [14], but in line with what was published in Denmark a decade ago [15] and older studies from the United States [16], the Netherlands [17], and Italy [18]. A recent study from Australia [19] evaluated adherence within 12 and 24 months of treatment and found that over half of those who attended a first follow-up visit did not attend a second. In our study, nearly 98% of the study population attended a first follow-up, and nearly 96% a second, but not necessarily within the recommended intervals.

Women treated in 2014 had theoretically 7–8 years of follow-up through to December 2021, and those treated in 2017 had at least 4 years. Two-thirds of women with incomplete follow-up had their most recent follow-up visit within 27 months of treatment. This means that more women neglect than schedule a follow-up. Incomplete follow-up was not associated with age or time between the last abnormal cytology result and the prior treatment (data not shown). We observed a higher proportion of women with incomplete follow-up in 2014–2017 than in women treated for CIN in 2006–2011 [14].

The relatively large proportion of women with incomplete follow-up in our analysis will lead to an underestimation in the cumulative incidence of residual disease, as this group likely contain cases of residual disease. Most studies on residual disease follow women for 2 years after treatment. However, to provide data for follow-up recommendations, studies with longer follow-up that distinguish between residual and recurrent CIN2+ are needed, as these risks will differ [14,20]. We found a non-significant increase in 5-year residual CIN2+ in the present study period (7.1% (95%, CI: 5.9–8.3)) relative to 2006–2011 (5.4% (95%, CI: 3.6–6.8)), which is consistent with a previous study that used a similar definition of residual and recurrent disease [20], and estimates from a meta-analysis of 24 studies with at least 18 months of post-treatment follow-up (6.6% (95%, CI: 4.9–8.4)) [21].

It is generally accepted that women treated for CIN should have a rigorous follow-up regimen due to the risk of residual disease, particularly cancer that was missed at treatment. This was demonstrated in this study, where six out of seven cancers were diagnosed within the first 15 months of treatment, all of which showed abnormal cytology and a positive HPV test at the first follow-up. Most countries recommend a first follow-up visit between 6 and 12 months post-treatment, and yearly follow-up visits thereafter until two [5] or three [4] negative co-tests are recorded. We found no CIN3+ among the 808 women with two consecutive negative co-tests after 1–6 years of follow-up. Two consecutive, negative co-tests are obviously very safe to rule out residual disease and short-term recurrent disease, but are very resource demanding for women (time, distress), clinics, and society (resources, costs).

Sweden recommends that women treated for CIN return to 3-year screening after a negative co-test at a first follow-up visit [2], while Denmark added the condition of free resection margins [6]. The data from the Kaiser Permanente Northern California (KPNC), which is used to inform the American Society of Colposcopy and Cervical Pathology (ASCCP) consensus guidelines [4], found that the 5-year risk of CIN3+ (1.7%) was too high to recommend immediate return to 3-year screening for women with a negative co-test at the first follow-up visit, and stated that an additional year of follow-up was required [22,23]. Two negative co-tests yielded a 5-year CIN3+ risk of 0.68% [22] which still indicates the need for a third negative co-test to attain risk estimates below 0.55%, which should be reached before returning women to 3-year screening [22,23].

In the present study, time was measured from the treatment to the event. As most women (>95%) had their first follow-up within 18 months of the treatment, the 5-year risk of CIN3+ was low, indicating that the second visit can be the first visit of the 3-year screening. As the sample size was small, we must be careful with conclusions regarding women with normal/ASCUS/ LSIL and HPV-negative results at first follow-up, but our data suggests that these women may return to 3-year screening. For women with all other cytology and HPV outcomes at the first follow-up, our data are in line with the 2019 recommendations from the ASCCP [21,22].

Partial genotyping may guide thresholds for follow-up in general and post-treatment surveillance [24]. However, data on the use of post-treatment partial genotyping in clinical decision-making are limited [24]. Setting thresholds for action will depend upon HPV prevalence, as will the number of unnecessary follow-up visits. The action thresholds applied for the 3-year CIN3+ risk is based on the 5-year CIN3+ risk by HPV prevalence among women with normal cytology [23]. As HPV infections wane over time, a first post-treatment follow-up visit closer to 12 months [25,26] may have an impact on the estimated action thresholds and the burden of follow-up visits. Taking partial genotyping into account, can we accept a higher action threshold for CIN3+ post-treatment than in the general screening population? Women with incomplete follow-up introduce selection bias in most studies, as their risk of CIN3+ is unknown [22,24]. A higher threshold for CIN3+ may simplify follow-up regimens and expand follow-up windows, making the program more acceptable to women that are “tired” of repeat colposcopy/biopsies followed by intermittent visits with co-tests without a definite, treatable diagnosis. As HPV16, 18, and 45 comprise 75% to 80% of the cervical cancer burden worldwide, these types may be useful for risk stratification in post-treatment follow-up guidelines [26].

The 2019 ASCCP follow-up guidelines are based on risk estimates for CIN3+ [4,22,23]. However, we do not know how they are practiced within the KPNC or elsewhere. This study demonstrates that Norwegian women and their physicians find it hard to adhere to the existing follow-up algorithm of 6 and 12 months post-treatment. It is difficult to communicate to women and their physicians the need to reach a 3-year risk for CIN-3+ post-treatment under 0.55% as the rationale for two follow-up visits within the first year or two of follow-up. We have demonstrated that the risk of CIN3+ is even lower at 3-year follow-up for women with normal cytology/ASCUS/LSIL and a negative HPV test at the first follow-up, a finding that challenges the existing guidelines on when to return to 3-year screening.

The strengths of the present study were the large, population-based sample and the long-term follow-up. Furthermore, we used firm definitions for residual and recurrent disease. The limitations include the retrospective study design and the relatively large proportion of women with incomplete follow-up, which probably does not differ from real-life surveillance in most countries.

## 5. Conclusions

Adherence to two follow-up visits at 6 and 12 months post-treatment, as outlined in the Norwegian follow-up guidelines, was low. This study underlines the importance of discriminating between residual and recurrent disease. The large proportion of women with incomplete follow-up is a challenge for the NCCSP and requires action. Despite this finding, the risk of CIN3+ among women with normal cytology/ASCUS/LSIL and a negative HPV test at the first follow-up is lower than 0.55%, which is the ASCCP threshold for when women may return to 3-year screening. For all other cytology/HPV outcomes, our findings support the 2019 ASCCP post-treatment follow-up guidelines.

## Figures and Tables

**Figure 1 ijerph-20-04739-f001:**
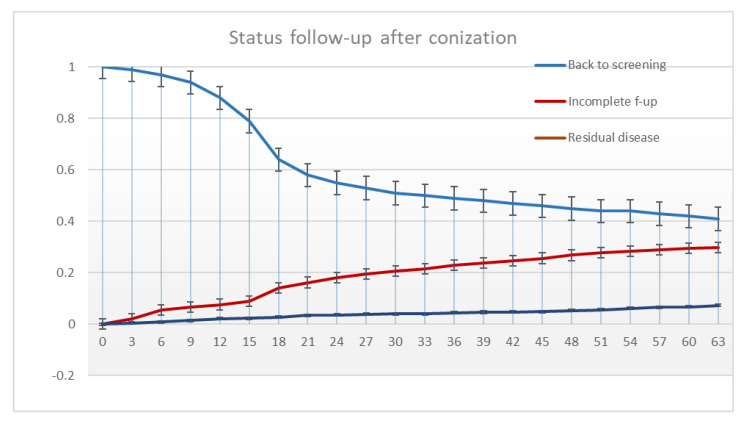
Time to residual disease (CIN2+), last observation for cases with incomplete follow-up, visit back to 3-year screening (One-minus cumulative proportion, survival analysis). CIN2+: CIN2, CIN3, and invasive cervical cancer.

**Table 1 ijerph-20-04739-t001:** Characteristics of the study population by treatment year.

	2014	2015	2016	2017	Total
	N = 286	N = 364	N = 352	N = 395	N = 1397
	%	%	%	%	%
Age					
19–24	7.0	3.6	2.8	4.3	4.3
25–29	26.9	33.0	27.8	34.7	30.9
30–33	14.3	15.4	15.1	18.2	15.9
34–49	37.8	35.7	37.5	28.4	34.5
50–69	13.3	11.5	15.6	12.9	13.3
70–85	0.7	0.8	1.1	1.5	1.1
Highest histology biopsy/cone					
Normal	0.7	1.1	1.1	0.8	0.9
CIN1	6.3	6.9	7.7	7.1	7.0
CIN2	47.2	51.9	56.3	53.2	52.4
CIN3	45.8	40.1	35.0	39.0	39.7
Resection margins					
Free	78.3	76.6	78.7	79.0	78.2
Not free	18.9	20.3	18.8	21.0	19.8
Inconclusive	2.8	3.0	2.6	0	2.0
Months from first abnormal cytology result					
≤12	44.4	49.2	49.7	51.6	49.0
13–42	21.0	21.2	24.1	24.8	22.9
43–78	5.6	8.5	4.5	7.3	6.6
79–321	29.0	21.2	21.6	16.2	21.5
Number of previous CIN treatment(s) (conization)					
0	94.1	93.1	96.0	96.5	95.0
1	5.6	6.0	4.0	3.5	4.7
2	0.3	0.8	0	0	0.3
Disease progression since most recent treatment					
Residual disease	1.7	2.7	3.4	2.5	2.6
Recurrent disease	4.1	4.1	0.6	1.0	2.4
HPV type/status					
Not examined	47.6	22.0	19.6	12.2	23.8
Negative	2.8	1.4	2.8	4.6	2.9
HPV16/HPV16 others	18.9	32.7	34.9	42.8	33.3
HPV18/not HPV16	3.8	6.9	8.2	8.1	6.9
Other HPV type/not HPV16/18	26.9	37.1	34.4	32.4	33.0

CIN1: cervical intraepithelial neoplasia grade 1, CIN2: cervical intraepithelial neoplasia grade 2, CIN3: cervical intraepithelial neoplasia grade 3, HPV: human papillomavirus.

**Table 2 ijerph-20-04739-t002:** Status of the study population at the end of follow-up (48 to 96 months after treatment).

Outcome		
	N	%
No follow-up	30	2.2
Incomplete follow-up	431	30.9
CIN2	76	5.4
CIN3	45	3.2
Squamous cell carcinoma	5	0.4
Adenocarcinoma	2	0.1
Back to 3-year screening	808	57.8
Total	1397	100.0

CIN2: cervical intraepithelial neoplasia grade 2, CIN3: cervical intraepithelial neoplasia grade 3.

**Table 3 ijerph-20-04739-t003:** Treatment and follow-up characteristics of the women with residual cervical cancer.

	At Conization			Follow-Up					Cervical Cancer	
ID	Age	Histo-Logy Results	Resection Margins	First Follow-Up		Number of Follow-Up Visits	Adherence to Follow-Up	Months to Diagnosis	Histology	Stage
				Cytology	HPV					
658894	27	CIN3	Not free	HSIL	18, other	10	Missed at reconization	40	Adeno-carcinoma	IB1
650766	31	CIN2	Free	HSIL	16, 18, other	5	As scheduled	14	Squamous- cell carcinoma	IA1
633739	39	CIN3	Free	HSIL	16	3	As scheduled	11	Squamous- cell carcinoma	IIA
614135	48 ^a^	CIN3	Free	HSIL	16	4	Missed at reconization	14	Squamous- cell carcinoma	IB2
586116	58 ^b^	CIN3	Not free	ASCUS	16	2	As scheduled	7	Adeno-carcinoma	IA1
570452	66 ^b^	CIN3	Not free	HSIL	Other	3	As scheduled	9	Squamous- cell carcinoma	IA1
570082	67 ^b^	CIN3	Not free	HSIL	16	2	As scheduled	7	Squamous- cell carcinoma	IB1

^a^ First conization 4 years before index conization; ^b^ incomplete participation in screening program. ASCUS: atypical squamous cells of undetermined significance, CIN2: cervical intraepithelial neoplasia grade 2, CIN3: cervical intraepithelial neoplasia grade 3, HPV: human papillomavirus, HSIL: high-grade squamous intraepithelial lesion.

**Table 4 ijerph-20-04739-t004:** Three- and 5-year risk of CIN3+ after treatment for CIN assessed by HPV and cytology outcomes at the first follow-up.

First Follow-Up				N Cases (≤60 Months after Treatment)		Cumulative Risk of CIN3+ (Residual Disease, Time from Treatment)	
HPV Result	Cytology Result	N	%	CIN2	CIN3+	3-Year % (95% CI)	5-Year % (95% CI)
Negative	Normal *	749	62.4	6	2	0.13 (0.0–0.38)	0.28 (0.0–0.67)
Negative	ASCUS/LSIL *	138	11.5	2	0	0.0	0.0
Negative	HSIL	14	1.2	1	2	14.3 (0.0–32.6)	14.3 (0.0–32.6)
Negative	All	901			4	0.33 (0.0–0.70)	0.45 (0.0–0.90)
Positive	Normal	42	3.5	3	1	2.4 (0.0–7.1)	2.4 (0.0–7.1)
Positive	ASCUS/LSIL	181	15.1	35	12	6.1 (2.6–9.6)	6.7 (3.0–10.4)
Positive	HSIL	77	6.4	15	26	29.9 (19.6–40.0)	33.8 (23.2–44.4)
	In total	1201	100	62	43		

* Normal/ASCUS/LSIL/HPV-negative: 36/60 months risk for CIN3+: 0.11 (0.0–0.33)/0.24 (0.0–0.57). ASCUS: atypical squamous cells of undetermined significance, CI: confidence interval, CIN: cervical intraepithelial neoplasia, CIN2: cervical intraepithelial neoplasia grade 2, CIN3+: cervical intraepithelial neoplasia grade 3 and invasive cervical cancer, HPV: human papillomavirus, HSIL: high-grade squamous intraepithelial lesion, LSIL: low-grade squamous intraepithelial lesion.

## Data Availability

The data presented in this study are available on request from the corresponding author.

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
