# Peer review of "May Women with a Negative Co-Test at First Follow-Up Visit Return to 3-Year Screening after Treatment for Cervical Intraepithelial Neoplasia?"

_ijerph, 2023, doi:10.3390/ijerph20064739_

Round 1

Reviewer 1 Report

I read with great interest the manuscript, which falls within the aim of this Journal. In my honest opinion, the topic is interesting enough to attract the readers’ attention. Nevertheless, the authors should clarify some points and improve the discussion, as suggested below.

Authors should consider the following recommendations:

-       Manuscript should be further revised in order to correct some typos and improve style.

-        Authors should discuss robust pieces of evidence about the use of new strategies for cervical cancer and pre-cancerous lesions screening and diagnosis, even using artificial intelligence and novel biomarkers (authors may refer to: PMID: 35742340).

-        I would recommend to stress novel pieces of evidence about high-risk HPV-negative high-grade cervical dysplasia, which seems to have more favorable outcomes than patients with documented high-risk-HPV infection (PMID: 33514481)

Author Response

R1_1:   I read with great interest the manuscript, which falls within the aim of this Journal. In my honest opinion, the topic is interesting enough to attract the readers’ attention. Nevertheless, the authors should clarify some points and improve the discussion, as suggested below.

Response: We are grateful for the overall assessment!

R1_2     Manuscript should be further revised in order to correct some typos and improve style.

Response: Before submission, the manuscript was edited and proofread twice by a professional international organization. We have in this revision checked extra all tables for subheadings and additional information as we found some typing errors.

R1_3    Authors should discuss robust pieces of evidence about the use of new strategies for cervical cancer and pre-cancerous lesions screening and diagnosis, even using artificial intelligence and novel biomarkers (authors may refer to: PMID: 35742340).

Response: We have with great interest read the editorial in JERPH (2022) “New advances in cervical cancer: from Bench to Bedside” (PMID: 35742340). The paper deals with minimal invasive surgery of cervical cancer, future outlook on cervical cancer therapy, combination therapy in cervical cancer, and molecular markers to predict prognosis.  Our paper deals with intervals for follow-up after treatment of premalignant lesions, which is not part of PMID 35742340.  We, therefore, find it unnatural to refer to the recommended paper in our manuscript.

R1_4    I would recommend to stress novel pieces of evidence about high-risk HPV-negative high-grade cervical dysplasia, which seems to have more favorable outcomes than patients with documented high-risk-HPV infection (PMID: 33514481)

Response: We are grateful for the opportunity to review the paper by Bogani G et al, entitled “High-risk HPV-positive and -negative high-grade cervical dysplasia: Analysis of 5-year outcomes” published in Gynecologic Oncology in 2021 (PMID: 33514481). The paper deals with the importance of recurrence of CIN3 by HPV-outcome prior to conization.  The paper has no information on HPV-outcome during follow-up and no data on follow-up intervals.  As our study assess the impact of outcomes of cytology and HPV-testing at 1st follow-up after treatment, the mentioned study had less relevant information for our study, and, thus, will be not included on our reference list.

Reviewer 2 Report

I read with great interest the Manuscript titled "May women with a negative co-test at first follow-up visit return to 3-year screening after treatment for cervical intraepithelial neoplasia?" which falls within the aim of the Journal.

In my honest opinion, the topic is interesting enough to attract the readers’ attention. 

Nevertheless, authors should clarify some point and improve the discussion citing relevant and novel key articles about the topic.

-The introduction should be extended and completed. I find interesting a reference to the efforts made for the prevention and early diagnosis of gynecological cancers (see PMID: 36141217).

-Inclusion/exclusion criteria should be better clarified by extending their description.

- Statistical analyses used should be described and justified.

- What are the actual clinical implications of this study? it is important to report the results obtained by the authors in the context of clinical practice and to adequately highlight what contribution this study adds to the literature already existing on the topic and to future study perspectives.

- Discussions can be expanded and improved by citing relevant articles (I suggest authors to read and insert in references the following article PMID: 33514481).

Considered all these points, I think it could be of interest for the readers and, in my opinion, it deserves the priority to be published after minor revisions.

Author Response

R2_1    I read with great interest the Manuscript titled "May women with a negative co-test at first follow-up visit return to 3-year screening after treatment for cervical intraepithelial neoplasia?" which falls within the aim of the Journal.    In my honest opinion, the topic is interesting enough to attract the readers’ attention.                                                                                                                                  Response: We are grateful for this positive feedback.

R2_2    The introduction should be extended and completed. I find interesting a reference to the efforts made for the prevention and early diagnosis of gynecological cancers (see PMID: 36141217). 

Response: The editorial “Advances on prevention and screening of gynecologic tumors: Are we stepping forward? by Giannini A et al in Healthcare in 2022 was an insightful overview of primary, secondary and tertiary prevention of gynecologic cancer.  The paper did not deal with follow-up regimen or screening tests applied in post-treatment testing of women having had surgical therapy for premalignant lesions.  As our study assess interval and 1st follow-up post-treatment testing, we find the recommended paper to be too general for our study. 

R2-3     Inclusion/exclusion criteria should be better clarified by extending their description.                     

Response: The 1st  paragraph, starting line 116, page 7, describes the only exclusioin criterion in this study.  “After exclusion of women with a diagnosis of cervical cancer in biopsies/cone specimens (n=27), 1397 women were included in our analyses”.  In our opinion this is clearly written.

In the results part we clearly describe how we have treated women with no follow-up (n=19 (1.4%)), just one follow-up, two follow-ups etc.  We have chosen to present follow-up results for all eligible women who were treated during the time-period of the study as this is a result, and not an exclusion criterion (page 9, 2nd paragraph, lines 176-177).

R2_3     Statistical analyses used should be described and justified.             Response: In page 8, 2nd paragraph before results, lines 150-151 we report the statistical analysis.

“All analyses were performed in SPSS version 27.0 with a Chi-square test, Fisher's exact test, and survival analyses. P-values <0.05 were considered statistically significant.” We agree that this description is very short.  However, we have explained how we have calculated time in 2nd paragraph, from top, page 8. To avoid necessary repetitions, we did not duplicate this information in the statistical part.

R2-4     What are the actual clinical implications of this study? it is important to report the results obtained by the authors in the context of clinical practice and to adequately highlight what contribution this study adds to the literature already existing on the topic and to future study perspectives.                                                                                                 

Response: According to guidelines in different countries, women are referred back to regular screening after two or three post-treatment visits with double negative outcomes of cytology and HPV-testing. Our study's results have significant implications for clinical practice by potentially reducing the number of follow-up visits required for women treated for CIN2+. This has the potential to alleviate the burden on both patients and the healthcare system, especially in countries with limited healthcare resources. Furthermore, our findings may also lead to improved patient outcomes by reducing anxiety and emotional distress during follow-up visits and lowering the risk of unnecessary procedures and associated complications. Overall, our study adds new information to the existing literature by providing evidence to support the safe return to 3-year screening after one visit with double negative test outcomes, thus potentially improving the efficiency and effectiveness of post-treatment follow-up for CIN2+."

R2-5     Discussions can be expanded and improved by citing relevant articles (I suggest authors to read and insert in references the following article PMID: 33514481).

 Response: (same response as to reviewer R1.4) : We are grateful for the opportunity to review the paper by Bogani G et al, entitled “High-risk HPV-positive and -negative high-grade cervical dysplasia: Analysis of 5-year outcomes” published in Gynecologic Oncology in 2021 (PMID: 33514481). The paper deals with the importance of recurrence of CIN3 by HPV-outcome prior to conization.  The paper has no information on HPV-outcome during follow-up and no data on follow-up intervals.  As our study assess the impact of outcomes of cytology and HPV-testing at 1st follow-up after treatment, the mentioned study had less relevant information for our study, and, thus, will be not included on our reference list.

R2-6     Considered all these points, I think it could be of interest for the readers and, in my opinion, it deserves the priority to be published after minor revisions.

Reviewer 3 Report

Reviewer Comments:

In this manuscript, the authors performed cross-sectional study comprised 1397 women, treated for CIN in Norway. Then they evaluated adherence to follow-up guidelines and assessed residual disease, using CIN3+ as the outcome. This report is meaningful because it provided the findings necessary to determine the effectiveness of post-treatment follow-up for CIN. However, there were still some aspects of the paper that were not fully described and discussed.

Specific critics are the following:

1) Are there any individual calls/recalls or other recommendations for follow-up visits for post-treatment cases of CIN in this study? Detailed description of recommendations for follow-up visits for post-treatment follow-up is required.

2) Detailed information should be provided on the medical institutions responsible for the first and second visits after CIN treatment and who is responsible for the cost of these visits.

3) The reasons why the risk of CIN3+ in cases with a negative first visit after treatment for CIN differs from that of ASCCP need to be examined more deeply, including differences in case background.

Author Response

In this manuscript, the authors performed cross-sectional study comprised 1397 women, treated for CIN in Norway. Then they evaluated adherence to follow-up guidelines and assessed residual disease, using CIN3+ as the outcome. This report is meaningful because it provided the findings necessary to determine the effectiveness of post-treatment follow-up for CIN. However, there were still some aspects of the paper that were not fully described and discussed.

Response: We are grateful for the reviewer's perspective of our study.

R3_1    Are there any individual calls/recalls or other recommendations for follow-up visits for post-treatment cases of CIN in this study? Detailed description of recommendations for follow-up visits for post-treatment follow-up is required.

Response: Following conization treatment for high-grade cervical intraepithelial neoplasia (CIN2+), a "test-of-cure" approach using cervical cytology and HPV testing is recommended at the 6-month and 12-month follow-up appointments. The first follow-up appointment is typically conducted by a gynecologist, with the second conducted by a general practitioner. To ensure adequate follow-up, the Clinical Pathology department at the University Hospital of North Norway (UNN) has implemented local reminder procedures to alert healthcare providers when a patient is overdue for a follow-up test. These reminders are typically sent to the last requisitioner, which is usually the gynecologist, despite the follow-up test being conducted by the general practitioner. It should be noted that Clinical Pathology at UNN does not send reminders directly to the patient, underscoring the importance of coordinated efforts between healthcare providers to ensure timely and appropriate follow-up care for this patient population. As this is a local procedure on recall, and not a national approach within our screening program, we have not added any information on this issue in the manuscript.

R3_2    Detailed information should be provided on the medical institutions responsible for the first and second visits after CIN treatment and who is responsible for the cost of these visits.

Response: The following text (pink) is added at the end of the 1st paragraph of M&M, page 6. “In 2014, the NCCSP recommended that women treated for CIN be followed up with co-tests at 6 and 12 months after treatment, free of charge, reimbursed by the public Norwegian social security  system, at the referral practitioner’s office either in general practice or in private gynecologic    settings (1).”

R3_3    The reasons why the risk of CIN3+ in cases with a negative first visit after treatment for CIN differs from that of ASCCP need to be examined more deeply, including differences in case background.

Response: This is not doable as we only have two cases of CIN3 among the 749 cases double negative at first follow-up. Furthermore, our 3- and 5-year incidences are lower than what is recommended for upper limit within the ASCCP guidelines (the upper CI for 5-year risk is above the 0.55% as stated by ASCCP).